# Vector-quantized Image Modeling with Improved VQGAN

**Jiahui Yu**    **Xin Li**    **Jing Yu Koh**    **Han Zhang**    **Ruoming Pang**    **James Qin**

**Alexander Ku**    **Yuanzhong Xu**    **Jason Baldridge**    **Yonghui Wu**

Google Research

`jiahuiyu@google.com`

## ABSTRACT

Pretraining language models with next-token prediction on massive text corpora has delivered phenomenal zero-shot, few-shot, transfer learning and multi-tasking capabilities on both generative and discriminative language tasks. Motivated by this success, we explore a Vector-quantized Image Modeling (**VIM**) approach that involves pretraining a Transformer to predict rasterized image tokens autoregressively. The discrete image tokens are encoded from a learned Vision-Transformer-based VQGAN (**ViT-VQGAN**). We first propose multiple improvements over vanilla VQGAN from architecture to codebook learning, yielding better efficiency and reconstruction fidelity. The improved ViT-VQGAN further improves vector-quantized image modeling tasks, including unconditional, class-conditioned image generation and unsupervised representation learning. When trained on ImageNet at $256 \times 256$ resolution, we achieve Inception Score (IS) of 175.1 and Fréchet Inception Distance (FID) of 4.17, a dramatic improvement over the vanilla VQGAN, which obtains 70.6 and 17.04 for IS and FID, respectively. Based on ViT-VQGAN and unsupervised pretraining, we further evaluate the pretrained Transformer by averaging intermediate features, similar to Image GPT (iGPT). This ImageNet-pretrained VIM-L significantly beats iGPT-L on linear-probe accuracy from 60.3% to 73.2% for a similar model size. ViM-L also outperforms iGPT-XL which is trained with extra web image data and larger model size.

## 1 INTRODUCTION

Natural language processing (NLP) has recently experienced dramatic improvements from learning general-purpose representations by pretraining language models on unlabeled text corpora. This strategy has produced large performance gains for a wide range of natural language generation (NLG) and natural language understanding (NLU) tasks (Dai & Le, 2015; Radford et al., 2018; 2019; Brown et al., 2020). Conceptually, generative pretraining models the data density $P(X)$ in a tractable way, with the hope of also helping discriminative tasks of $P(Y|X)$ (Lasserre et al., 2006); importantly, there are no limitations on whether the signals are from the language domain or others, such as vision.

In computer vision, in contrast, most recent unsupervised or self-supervised learning research focuses on applying different random augmentations to images, with the pretraining objective to distinguish image instances (Chen et al., 2020b; He et al., 2020; Chen et al., 2020d; Grill et al., 2020; Chen et al., 2020c; Caron et al., 2021). The quality of learned representation relies on manually chosen augmentations, such as random brightness, cropping, blurring, and others. Chen et al. (2020a) explored GPT-style (Radford et al., 2018) generative pretraining on images by autoregressively predicting pixels without incorporating knowledge of the 2D structure. Each pixel is represented as a 9-bit value created by clustering (R, G, B) pixel values, using k-means with k=512. Unfortunately, this color encoding does not scale to typical image resolutions as it entails very long sequences to represent the image (*e.g.*, $224 \times 224$ resolution leads to 50,176 tokens per image), and this demands much more memory and computation for training, compared to language models. As a result, iGPT's maximum resolution is $64 \times 64$ for image recognition at scale—which severely limits its representation capabilities.

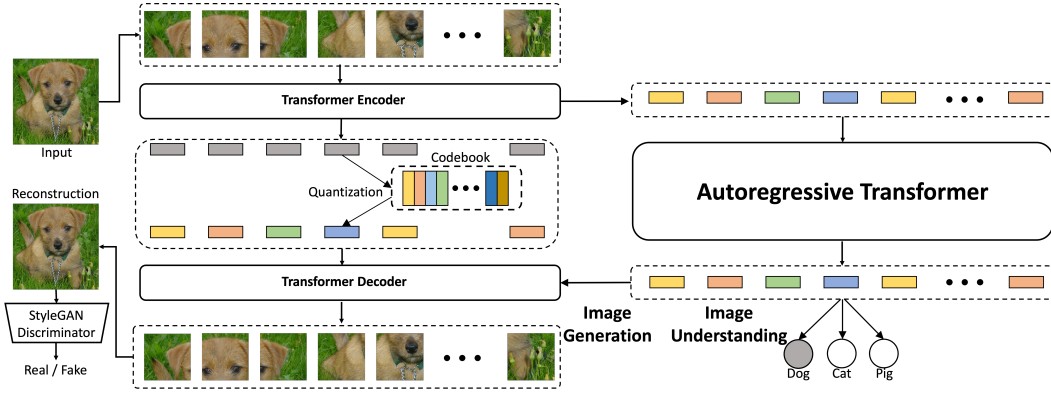

Figure 1: Overview of ViT-VQGAN (left) and Vector-quantized Image Modeling (right) for both image generation and image understanding.

Remarkable image generation results have been achieved by pre-quantizing images into discrete latent variables and modeling them autoregressively, including VQVAE (Oord et al., 2017), DALL-E (Ramesh et al., 2021) and VQGAN (Esser et al., 2021). In these approaches, a convolution neural network (CNN) is learned to auto-encode an image and a second stage CNN or Transformer is learned to model the density of encoded latent variables. These have been proved effective for image generation, but few studies have evaluated the learned representation in discriminative tasks (Ramesh et al., 2021; Esser et al., 2021).

We explore an approach we refer to as Vector-quantized Image Modeling (VIM) and apply it to both image generation and image understanding tasks. VIM follows a two-stage approach:

- **Stage 1: Image Quantization.** Given an image of resolution $256 \times 256$, a Vision-Transformer-based VQGAN encodes it into $32 \times 32$ discretized latent codes where the codebook size is $8192$. We propose multiple improvements–from architecture to codebook learning–to VQGAN (Esser et al., 2021). The resulting ViT-VQGAN is more efficient and improves reconstruction fidelity in terms of pixel-wise reconstruction metrics, Inception Score (IS) and Fréchet Inception Distance (FID). ViT-VQGAN is trained end-to-end on image-only data with combined objective functions of logit-laplace loss, $\ell_2$ loss, adversarial loss and perceptual loss (Johnson et al., 2016; Zhang et al., 2018).

- **Stage 2: Vector-quantized Image Modeling.** We train a Transformer model to predict rasterized $32 \times 32 = 1024$ image tokens autoregressively, where image tokens are encoded by a learned Stage 1 ViT-VQGAN. For unconditional image synthesis or unsupervised learning, we pretrain a decoder-only Transformer model to predict the next token. For class-conditioned image synthesis, a class-id token is prepended before the image tokens. To evaluate the quality of unsupervised learning, we average the intermediate Transformer features and learn a linear head to predict the logit of the classes (*a.k.a.*, linear-probe).

We show that one key component for improving both image generation and image understanding with VIM is to have a better image quantizer with respect to both computational efficiency and reconstruction quality. An efficient quantizer can speed up Stage 2 training, where random augmentations are applied first to an image, followed by the encoder of image quantizer to obtain the input tokens. Moreover, an image quantizer with better reconstruction quality can reduce information loss compared with the original image in pixel space, which is critical for image understanding tasks.

The evaluations of our proposed ViT-VQGAN and VIM are studied with three aspects. (1) We evaluate the image quantizer based on reconstruction quality metrics including $\ell_1$ distance, $\ell_2$ distance, log-laplace distance, as well as Inception Score (IS) and Fréchet Inception Distance (FID) of reconstructed images. (2) We evaluate the capabilities of the learned quantizer for unconditional or class-conditioned image synthesis based on FID and IS, and compare with other methods. (3) We rely on linear-probe accuracy to evaluate representations with the common intuition that good features should linearly separate the classes of downstream tasks.

## 2 RELATED WORK

**Image Synthesis.** Image generation has received much attention with the progress of deep generative models, including Generative Adversarial Networks (GANs) (Goodfellow et al., 2014; Zhang et al., 2019b), Variational Autoencoders (VAEs) (Kingma & Welling, 2014; Vahdat & Kautz, 2020), Diffusion Models (Song & Ermon, 2019; Dhariwal & Nichol, 2021) and Autoregressive Models (van den Oord et al., 2016; Parmar et al., 2018). Unlike many autogressive methods which generate sequence directly in pixel space, VQVAE (van den Oord et al., 2017; Razavi et al., 2019) decomposes the image generation process into two stages: the first stage trains a vector quantized autoencoder with image reconstruction objective to convert an image into a shorter sequence of discrete tokens. Then the second stage learns an autoregressive model, e.g., PixelSNAIL (Chen et al., 2018), to model the underlying distribution of token sequences. Driven by the effectiveness of VQVAE and progress in sequence modeling (Vaswani et al., 2017; Devlin et al., 2019), many approaches follow the two-stage paradigm. DALL-E (Ramesh et al., 2021) improves token prediction in second stage by using Transformers (Vaswani et al., 2017), resulting in a strong text-to-image synthesis model. VQGAN (Esser et al., 2021) further uses adversarial loss and perceptual loss (Johnson et al., 2016; Zhang et al., 2018) to train a better autoencoder in the first stage to synthesize greater detail in images.

**Image Recognition with Generative Pretraining.** Many image generation models (Goodfellow et al., 2014; Kingma & Welling, 2014; Radford et al., 2016; Donahue et al., 2017; Higgins et al., 2017) have been studied for their capabilities in representation learning. However, their performance is usually not superior to competing self-supervised approaches that solve auxiliary classification tasks (Noroozi & Favaro, 2016a; Gidaris et al., 2018a; van den Oord et al., 2018). BigBiGAN (Donahue & Simonyan, 2019a) first demonstrated that a generation-based model can match other self-supervised methods in representation learning on ImageNet. iGPT (Chen et al., 2020a) uses the autoregressive objective to learn a giant transformer that directly predicts pixel values, producing even more competitive results. Compared to iGPT, our method first tokenizes the original image into discrete image tokens and then trains a transformer to predict them. As a result, our approach obtains comparable results with smaller model and less data. Similar to our method in predicting image tokens, BEiT (Bao et al., 2021) follows pre-training scheme of BERT Devlin et al. (2019) by learning to recover randomly masked image tokens with a bidirectional transformer. Unlike BEiT, we explore vector-quantized image modeling for image generation in addition to image recognition.

## 3 VECTOR-QUANTIZED IMAGES WITH VIT-VQGAN

The Vector-quantized Variational AutoEncoder (VQVAE) (van den Oord et al., 2017) is a CNN-based auto-encoder whose latent space is a matrix of discrete learnable variables, trained end-to-end via straight-through estimation. Esser et al. (2021) introduce VQGAN, a model which improves upon VQVAE by introducing an adversarial loss produced by a discriminator. Below, we introduce further improvements to VQGAN that boost efficiency and enhance reconstruction quality.

### 3.1 VQGAN WITH VISION TRANSFORMERS

The core network architectures used by both VQVAE and VQGAN to encode and reconstruct images are CNNs. VQGAN introduces transformer-like elements in the form of non-local attention block (Zhang et al., 2019a), allowing it to capture distant interactions with fewer layers. We propose taking this approach one step further by replacing the CNN encoder and decoder with Vision Transformer (ViT) (Dosovitskiy et al., 2020), as shown in Figure 1. Given sufficient data (for which unlabeled image data is plentiful) we find that ViT-VQGAN is less constrained by the inductive priors imposed by convolutions. Furthermore, ViT-VQGAN yields better computational efficiency on accelerators, and produces higher quality reconstructions, as shown in Table 1.

The encoder of ViT-VQGAN first maps $8{\times}8$ non-overlapping image patches into image tokens, followed by Transformer blocks, encoding a $256{\times}256$ resolution image into a $32{\times}32{=}1024$ token sequence. The decoder performs the inverse operation, mapping each image token from latent variables back to $8 \times 8$ image patches and regrouping them into a $256 \times 256$ image (see Figure 1). At the output of transformer blocks, we apply a two-layer feed-forward network with a tanh activation layer

| Architecture | Model Size (encoder-decoder) | Throughput ↑ (imgs/sec) | $\ell_2$ loss ↓ (1e-2) | Logit-Laplace loss ↓ | FID ↓ | IS ↑ |
|---|---|---|---|---|---|---|
| ViT-VQGAN | Small-Small | **1520** | 3.34 | -2.44 | 1.99 | 184.4 |
| CNN-VQGAN | Channels × 1 | 946 | 3.81 | -2.36 | 2.26 | 178.7 |
| ViT-VQGAN | Base-Base | 960 | **3.09** | **-2.54** | **1.55** | **190.2** |
| CNN-VQGAN | Channels × 2 | 400 | 3.44 | -2.46 | 1.91 | 183.4 |
| ViT-VQGAN | Small-Large | 384 | **2.88** | **-2.58** | **1.28** | **192.3** |

Table 1: ViT-VQGAN achieves better speed-quality trade-offs compared with CNN-VQGAN. This in turn further speeds up Stage 2 training. Throughputs are benchmarked with the same 128 CloudTPUv4 devices.

in the middle. No activation is applied at the output of ViT-VQGAN encoder or decoder. We find that this simple approach yields high quality reconstructions without any noticeable grid artifacts.

## 3.2 CODEBOOK LEARNING

Vanilla VQVAEs usually suffer from low codebook usage due to the poor initialization of the codebook. Therefore, during training a significant portion of codes are rarely used, or *dead*. The reduction in effective codebook size results in worse reconstructions in stage 1 quantizer training and poor diversity in stage 2 for image synthesis. As a result, VQGAN (Esser et al., 2021) relies on top-$k$ and top-$p$ (nucleus) sampling heuristics (Holtzman et al., 2020) with a default codebook size of 1024 to obtain best results for image synthesis. We propose two improvements that can significantly encourage the codebook usage even with a larger codebook size of 8192. During image synthesis, we perform simple sampling with temperature of 1.0 without top-$k$ and top-$p$ heuristics.

The training objective of vector-quantization is defined as follows:

$$L_{\text{VQ}} = \|\text{sg}[z_e(x)] - e\|_2^2 + \beta\|z_e(x) - \text{sg}[e]\|_2^2. \tag{1}$$

Here, $\text{sg}(x) \equiv x, \frac{\mathrm{d}}{\mathrm{d}x}\text{sg}(x) \equiv 0$ is the stop-gradient operator, $\beta$ is a commitment loss hyperparameter set to 0.25 in all our experiments, and $e$ is the codebook vector. The quantized codebook index is determined by looking up the codebook vector closest to the input features $z_e(x)$ in terms of the Euclidean distance, yielding $i = \text{argmin}_j \|z_e(x) - e_j\|_2^2$.

**Factorized codes.** We introduce a linear projection from the output of the encoder to a low-dimensional latent variable space for code index lookup (*e.g.*, reduced from a 768-d vector to a 32-d or 8-d vector per code) and find it has an immediate boost of codebook usage. The factorization can be viewed as decoupling code lookup and code embedding: we lookup the the closest variable encoded from input on a lower-dimensional lookup space and then project the matched latent code to the high-dimensional embedding space. Our experiments show reducing dimension of lookup space from 256-d to 32-d consistently improves reconstruction quality. A detailed illustration is provided in the supplementary materials.

**$\ell_2$-normalized codes.** We also apply $\ell_2$ normalization on the encoded latent variables $z_e(x)$ and codebook latent variables $e$. The codebook variables are initialized from a normal distribution. By mapping all latent variables on a sphere, the Euclidean distance of $\ell_2$-normalized latent variables $\|\ell_2(z_e(x)) - \ell_2(e_j)\|_2^2$ evolves to the cosine similarity of two vectors between $z_e(x)$ and $e$, further improving training stability and reconstruction quality shown in our experiments.

## 3.3 VIT-VQGAN TRAINING LOSSES

We use a combination of logit-laplace loss, $\ell_2$ loss, perceptual loss (Johnson et al., 2016; Zhang et al., 2018) based on VGG network (Simonyan & Zisserman, 2014) and GAN loss with architecture of StyleGAN discriminator (Karras et al., 2020). Loss balancing weights are configured with a hyper-parameter sweep to optimize image reconstruction quality, codebook usage, FID and Inception Score. After the sweep, we apply the same set of hyper-parameters of training losses to all datasets including CelebA-HQ, FFHQ, and ImageNet. Logit-Laplace loss can be viewed as normalized $\ell_1$ loss which assumes the noise at the pixel level is laplace-distributed while $\ell_2$ loss assumes

| Model | Size | #Params | #Blocks | #Heads | Model Dim | Hidden Dim | Dropout | #Tokens |
|-------|------|---------|---------|--------|-----------|------------|---------|---------|
| ViT-VQGAN | Small | 32M | 8 | 8 | 512 | 2048 | 0.0 | 1024 |
| ViT-VQGAN | Base | 91M | 12 | 12 | 768 | 3072 | 0.0 | 1024 |
| ViT-VQGAN | Large | 599M | 32 | 16 | 1280 | 5120 | 0.0 | 1024 |
| VIM | Base | 650M | 24 | 16 | 1536 | 6144 | 0.1 | 1024 |
| VIM | Large | 1697M | 36 | 32 | 2048 | 8192 | 0.1 | 1024 |

Table 2: Transformer architectures of Stage 1 ViT-VQGAN and Stage 2 VIM.

the noise is of a Gaussian distribution. We find logit-laplace loss contributes to codebook usage while $\ell_2$ loss and perceptual loss significantly contribute to FID. The final loss combination we used by default is $L = L_{\text{VQ}} + 0.1\,L_{\text{Adv}} + 0.1\,L_{\text{Perceptual}} + 0.1\,L_{\text{Logit-laplace}} + 1.0L_2$.

One caveat on the VGG-based perceptual loss is that the VGG network is pretrained with supervised classification loss, so the supervision might *leak* into Stage 2 for linear-probe accuracy measurement. Thus, for all of our reported unsupervised learning results, we exclude the perceptual loss during ViT-VQGAN training. For all unconditional and class-conditioned image synthesis, we use ViT-VQGAN quantizers trained with perceptual loss, as it leads to higher-fidelity reconstructions.

## 4 Vector-quantized Image Modeling

With a learned ViT-VQGAN, images are encoded into discrete latent code ids flattened in the raster order, similar to Image GPT (Chen et al., 2020a). A decoder-only Transformer model is used to model the density of image data $P(x)$ autoregressively as

$$P(x) = \prod_{i=1}^{n} P(x_i | x_1, x_2, ..., x_{i-1}; \theta), \qquad (2)$$

where $\theta$ is learnable weights. The training objective is to minimize the negative log-likelihood of the data $L = \mathbb{E}_{x \in X}[-log P(x)]$.

Table 2 summarizes the architecture configurations for the Transformers. We first embed discrete image token ids into a learnable embedding space at each position, with an additive learnable 2D positional embedding. Both embedding dimensions are the same as model dimension. We apply a stack of Transformer blocks to the inputs with causal attention over the entire sequence. A dropout ratio of 0.1 is used in all residual, activation and attention outputs. At the final layer of all Transformer blocks, we apply an additional layer normalization.

### 4.1 Image Synthesis

With a pretrained generative Transformer model, unconditional image generation is achieved by simply sampling token-by-token from the output softmax distribution. All samples used for both qualitative and quantitative results are obtained without temperature reduction. The sampled tokens are then fed into the decoder of ViT-VQGAN to decode output images. Our default Stage 1 ViT-VQGAN encodes input images of resolution $256 \times 256$ into $32 \times 32$ latent codes with a codebook size 8192, while Stage 2 Transformer takes the flattened image tokens with total a length of 1024.

Class-conditioned ImageNet generation is also a widely used benchmark for measuring capabiltiy of models for image synthesis. We extend the unconditional generation to class-conditioned generation by prepending a class-id token before the image tokens. Separate embedding layers are learned from scratch for class-id token and image tokens, with the embedding dimension the same as the Transformer model dimension. During sampling, a class-id token is provided at the first position to decode the remaining image tokens autoregressively.

### 4.2 Unsupervised Learning

For the image understanding task, we feed all image tokens of the input into a pretrained Transformer, and get a sequence of 1024 token features. Similar to Image GPT (Chen et al., 2020a),

| Model | Dataset | Latent Size | dim $\mathcal{Z}$ | FID on Validation |
|---|---|---|---|---|
| DALL-E dVAE | Web data | $32 \times 32$ | 8192 | 32.00 |
| VQGAN | ImageNet | $16 \times 16$ | 1024 | 7.94 |
| VQGAN | ImageNet | $16 \times 16$ | 16384 | 4.98 |
| VQGAN* | ImageNet | $32 \times 32$ | 8192 | 1.49 |
| VQGAN** | ImageNet | $64 \times 64$ & $32 \times 32$ | 512 | 1.45 |
| ViT-VQGAN (Ours) | ImageNet | $32 \times 32$ | 8192 | 1.28 |
| ViT-VQGAN (Ours) | CelebA-HQ | $32 \times 32$ | 8192 | 4.66 |
| ViT-VQGAN (Ours) | FFHQ | $32 \times 32$ | 8192 | 3.13 |

Table 3: Fréchet Inception Distance (FID) between reconstructed validation split and original validation split on ImageNet, CelebA-HQ and FFHQ. * denotes models trained with Gumbel-Softmax reparameterization as in Ramesh et al. (2021). ** denotes models trained with multi-scale hierarchical codebook as in Razavi et al. (2019).

we take a layer output at a specific block $l$ over total blocks $L$, average over the sequence of token features (frozen) and insert a softmax layer (learnable) projecting averaged feature to class logits. We only take one specific Transformer block output instead of concatenating different block outputs as in iGPT (Chen et al., 2020a). We find that most discriminating feature for the linear-probe is typically near the middle of all Transformer blocks.

## 5 EXPERIMENTS

### 5.1 IMAGE QUANTIZATION

We train the proposed ViT-VQGAN on three datasets separately, CelebA-HQ (Karras et al., 2019), FFHQ (Karras et al., 2019), and ImageNet (Krizhevsky et al., 2012). For CelebA-HQ and FFHQ, we follow the default train and validation split as VQGAN (Esser et al., 2021).[1] For Stage 1 image quantization, three different architecture sizes are experimented, Small, Base and Large for either encoder or decoder, as defined in Table 2. The smallest ViT-VQGAN-SS has a Small-size encoder and Small-size decoder, while ViT-VQGAN-BB has a Base-size encoder and Base-size decoder. The largest ViT-VQGAN-SL has an asymmetric Small-size encoder and Large-size decoder, with the motivation that Stage 2 training only requires forward propagation of the encoder of ViT-VQGAN (in inference/decoding for image synthesis, the decoder of ViT-VQGAN is still required to decode images from codes predicted during Stage 2).

We train all ViT-VQGAN models with a training batch size of 256 distributed across 128 CloudT-PUv4 for a total 500,000 training steps. For both ViT-VQGAN and StyleGAN discriminator, Adam optimizer (Kingma & Ba, 2014) is used with $\beta_1 = 0.9$ and $\beta_2 = 0.99$ with the learning rate linearly warming up to a peak value of $1 \times 10^{-4}$ over 50,000 steps and then decaying to $5 \times 10^{-5}$ over the remaining 450,000 steps with a cosine schedule. We use a decoupled weight decay (Loshchilov & Hutter, 2017) of $1 \times 10^{-4}$ for both ViT-VQGAN and StyleGAN discriminator. All models are trained with an input image resolution $256 \times 256$ on CloudTPUv4.

Table 3 shows FID between reconstructed images and original images in the validation split on ImageNet, CelebA-HQ and FFHQ datasets. Without multi-scale hierarchical codebook or gumbel-softmax, ViT-VQGAN is able to achieve better FID with a large codebook size of $8192$ compared with vanilla VQGAN.

Table 4 provides extensive ablations on the proposed modifications, with empirical results on mean $\ell_1$ distance, $\ell_2$ distance, logit-laplace distance, Inception Score and FID on ImageNet. Among different model sizes, ViT-VQGAN-SS (small-encoder, small-decoder) performs worse than ViT-VQGAN-BB (base-encoder, base-decoder) and ViT-VQGAN-SL (small-encoder, large-decoder), but achieves much better throughput. The CNN-based VQGAN architecture is worse in both quality and throughput compared with ViT-based VQGAN. The StyleGAN-based discriminator (Karras et al., 2019) is more stable and yields better reconstruction quality than PatchGAN (Isola et al., 2017)

---

[1]https://github.com/CompVis/taming-transformers

| Ablation on | Encoder Size | Decoder Size | Architecture | Discriminator | Latent dim | $\ell_2$-normalized | $\ell_1$ (1e-2) ↓ | $\ell_2$ (1e-3) ↓ | Logit-Laplace ↓ | IS ↑ | FID ↓ | Codebook Usage ↓ | Throughput ↓ |
|---|---|---|---|---|---|---|---|---|---|---|---|---|---|
| | Base | Base | ViT | StyleGAN | 32 | ✓ | 3.06 | 3.09 | -2.54 | 190.2 | 1.55 | 96% | 960 |
| Model Size | Small | Small | ViT | StyleGAN | 32 | ✓ | 3.22 | 3.34 | -2.44 | 184.4 | 1.99 | 95% | 1520 |
| | Small | Large | ViT | StyleGAN | 32 | ✓ | 2.93 | 2.88 | -2.58 | 192.3 | 1.28 | 95% | 384 |
| Architecture | - | - | CNN | StyleGAN | 32 | ✓ | 3.45 | 3.81 | -2.36 | 178.7 | 2.26 | 63% | 946 |
| | Base | Base | ViT | PatchGAN | 32 | ✓ | 2.82 | 2.58 | -2.62 | 165.6 | 3.88 | 89% | 1227 |
| Codebook Learning | Base | Base | ViT | StyleGAN | 256 | ✓ | 3.60 | 4.28 | -2.38 | 160.1 | 3.68 | 4% | 954 |
| | Base | Base | ViT | StyleGAN | 128 | ✓ | 3.41 | 3.93 | -2.44 | 173.9 | 2.77 | 14% | 960 |
| | Base | Base | ViT | StyleGAN | 64 | ✓ | 3.18 | 3.37 | -2.49 | 179.5 | 2.50 | 37% | 960 |
| | Base | Base | ViT | StyleGAN | 16 | ✓ | 3.00 | 2.96 | -2.54 | 191.2 | 1.50 | 95% | 960 |
| | Base | Base | ViT | StyleGAN | 8 | ✓ | 2.98 | 2.92 | -2.55 | 189.5 | 1.52 | 96% | 960 |
| | Base | Base | ViT | StyleGAN | 4 | ✓ | 3.55 | 4.18 | -2.37 | 143.8 | 3.68 | 96% | 960 |
| | Base | Base | ViT | StyleGAN | 32 | ✗ | 4.13 | 5.41 | -2.20 | 123.6 | 5.44 | 2% | 960 |

Table 4: Ablation study on ViT-VQGAN. The codebook usage is calculated as the percentage of used codes given a batch of 256 test images averaged over the entire test set.

| CelebA-HQ $256 \times 256$ | | FFHQ $256 \times 256$ | |
|---|---|---|---|
| Method | FID ↓ | Method | FID ↓ |
| GLOW (Kingma & Dhariwal, 2018) | 69.0 | VDVAE ($t = 0.7$) (Child, 2021) | 38.8 |
| NVAE (Vahdat & Kautz, 2020) | 40.3 | VDVAE ($t = 1.0$) | 33.5 |
| PIONEER (Heljakka et al., 2018) | 25.3 | VDVAE ($t = 0.8$) | 29.8 |
| NCPVAE (Aneja et al., 2020) | 24.8 | VDVAE ($t = 0.9$) | 28.5 |
| VAEBM (Xiao et al., 2021) | 20.4 | *VQGAN*+P.SNAIL | 21.9 |
| Style ALAE (Pidhorskyi et al., 2020) | 19.2 | BigGAN | 12.4 |
| DC-VAE (Parmar et al., 2021) | 15.8 | U-Net GAN (Schonfeld et al., 2020) | 10.9 |
| PGGAN (Karras et al., 2018) | 8.0 | StyleGAN2 (Karras et al., 2020) | 3.8 |
| VQGAN (w/ top-$k$ sampling) | 10.2 | VQGAN (w/ top-$k$ sampling) | 9.6 |
| **ViT-VQGAN (Ours)** | 7.0 | **ViT-VQGAN (Ours)** | 5.3 |

Table 5: FID comparison with unconditional image synthesis on CelebA-HQ and FFHQ.

(which was used for VQGAN). For codebook learning, factorized codes with low-dimensional latent variables consistently achieve better reconstruction quality when the latent dimension is reduced from 256 to 16 or 8. Moreover, removing $\ell_2$-normalization leads to much worse results.

## 5.2 IMAGE SYNTHESIS

On top of the pre-learned ViT-VQGAN image quantizer, we train stage 2 transformer models for unconditional and class-conditioned image synthesis and compare with previous work. We use a default model size of ViT-VQGAN-SS (small-encoder, small-decoder) for stage 1 and VIM-Large for stage 2 (model architectures are listed in Table 2), as we find it beneficial to put more computation in stage 2 while keeping stage 1 transformers lightweight. We also present a model size ablation study and comparison with VQGAN in the Appendix. Models are trained with a global training batch size of 1024 for a total of $450,000$ training steps. We use Adam optimizer (Kingma & Ba, 2014) with $\beta_1 = 0.9$ and $\beta_2 = 0.96$ with the learning rate linearly warming up to a peak constant value of $4.5 \times 10^{-4}$ over the first 5000 steps and then exponentially decaying to $1 \times 10^{-5}$ starting from 80,000 steps. To save memory, we use a factorized version of Adam, Adafactor (Shazeer & Stern, 2018), with the first moment quantized into Int8 and factorized second moments. No other techniques like mixed-precision training, model sharding, or gradient compression is used. All models are trained with an input image resolution $256 \times 256$ on CloudTPUv4.

| Model | Acceptance Rate | FID | IS |
|---|---|---|---|
| Validation data | 1.0 | 1.62 | 235.0 |
| DCTransformer (Nash et al., 2021) | 1.0 | 36.5 | n/a |
| BigGAN (Brock et al., 2019) | 1.0 | 7.53 | 168.6 |
| BigGAN-deep | 1.0 | 6.84 | 203.6 |
| IDDPM (Nichol & Dhariwal, 2021) | 1.0 | 12.3 | n/a |
| ADM-G, no guid. (Dhariwal & Nichol, 2021) | 1.0 | 10.94 | 101.0 |
| ADM-G, 1.0 guid. | 1.0 | 4.59 | 186.7 |
| VQVAE-2 (Razavi et al., 2019) | 1.0 | ∼31 | ∼45 |
| VQGAN (Esser et al., 2021) | 1.0 | 17.04 | 70.6 |
| VQGAN | 0.5 | 10.26 | 125.5 |
| VQGAN | 0.25 | 7.35 | 188.6 |
| **ViT-VQGAN (Ours)** | 1.0 | 4.17 | 175.1 |
| **ViT-VQGAN (Ours)** | 0.5 | 3.04 | 227.4 |

Table 6: FID comparison for class-conditional image synthesis on ImageNet with resolution $256 \times 256$. Acceptance rate shows results based on ResNet-101 classifier-based rejection sampling.

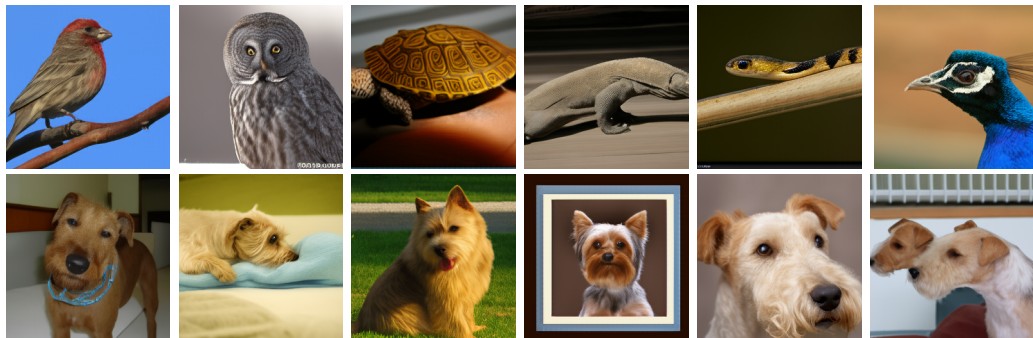

Figure 2: Uncurated set of samples from class-conditioned image generation trained on ImageNet. Top row shows sampled class ids while bottom row shows fine-grained dog species from class id 184 to 189. More samples are shown in Appendix.

Our main results on unconditional image synthesis on CelebA-HQ and FFHQ are summarized in Table 5. Without top-$k$ and top-$p$ (nucleus) sampling heuristics, we achieve FID of 7.0 on CelebA-HQ and 5.3 on FFHQ, significantly better than VQGAN (Esser et al., 2021). Table 6 shows class-conditioned image synthesis models on ImageNet, following Section 4.1. Based on ViT-VQGAN-SS, we achieve IS of 175.1 and FID of 4.17, improving over the IS of 70.6 and FID of 17.04 with vanilla VQGAN. When applied with classifier-based rejection sampling, the ViT-VQGAN based model further achieves best FID of 3.04 and best Inception Score of 321.7. Qualitative results are sampled and shown in Figure 2 (see the Appendix for more).

## 5.3 UNSUPERVISED LEARNING

After the generative pretraining to autoregressively model the density of ViT-VQGAN quantized image tokens, we evaluate the learned representation under the common linear protocol on ImageNet classification. We follow the same training hyper-parameters as the unconditional image synthesis models on ImageNet, and use ViT-VQGAN-SS image quantizer for better training throughput. As discussed in Section 3.3, the ViT-VQGAN-SS image quantizer is trained without perceptual loss for unsupervised learning (perceptual loss is based on a supervised VGG network trained on ImageNet, which may make comparison unfair). We apply an average pooling over the token features at a specific transformer block $l$ from totally $L$ blocks. Similar to findings reported in iGPT (Chen et al., 2020a), the representations from the middle transformer blocks has better linear-probe accuracy (more study can be found in Appendix). Specifically, we use the Transformer block of index 15 (36 blocks in total) for VIM-Large and index 10 (24 blocks in total) for VIM-Base (architecture configurations are listed in Table 2).

|  | Method | #Tokens | Features | Params | Top-1 ↑ |
|---|---|---|---|---|---|
| Discriminative Pretraining | Jigsaw (Noroozi & Favaro, 2016b) | - | 4096 | 94M | 44.6 |
|  | RelativePosition (Doersch et al., 2015) | - | 4096 | 94M | 51.4 |
|  | Rotation (Gidaris et al., 2018b) | - | 8192 | 86M | 55.4 |
|  | AMDIM (Bachman et al., 2019) | - | 8192 | 626M | 68.1 |
|  | CPC v2 (Henaff, 2020) | - | 4096 | 303M | 71.5 |
|  | MoCo (He et al., 2020) | - | 8192 | 375M | 68.6 |
|  | SimCLR (Chen et al., 2020b) | - | 8192 | 375M | 76.5 |
|  | SwAV (Caron et al., 2020) | - | 2048 | 93M | 75.3 |
|  | DINO (Caron et al., 2021) | - | 2048 | 85M | 75.3 |
|  | BYOL (Grill et al., 2020) | - | 8192 | 375M | 78.6 |
| Generative Pretraining | BiGAN (Donahue et al., 2016) | - | 512 | 138M | 31.0 |
|  | BigBiGAN (Donahue & Simonyan, 2019b) | - | 4096 | 86M | 56.6 |
|  | BigBiGAN | - | 16384 | 344M | 61.3 |
|  | iGPT-L (Chen et al., 2020a) | $32 \times 32$ | 1536 | 1362M | 60.3 |
|  | iGPT-L | $48 \times 48$ | 1536 | 1362M | 65.2 |
|  | iGPT-XL (extra data) | $64 \times 64$ | 3072 | 6801M | 68.7 |
|  | iGPT-XL (extra data, feature ensemble) | $64 \times 64$ | $5 \times 3072$ | 6801M | 72.0 |
|  | **VIM-Base (Ours)** | $32 \times 32$ | 1024 | 650M | 65.1 |
|  | **VIM-Large (Ours)** | $32 \times 32$ | 2048 | 1697M | 73.2 |
|  | VIM-Base + DALL-E dVAE quantizer | $32 \times 32$ | 1024 | 650M | 63.8 (-1.3) |
|  | VIM-Base + CNN-VQGAN quantizer | $32 \times 32$ | 1024 | 650M | 61.8 (-3.3) |

Table 7: Linear-probe accuracy with different unsupervised learning methods on ImageNet. DALL-E dVAE (Ramesh et al., 2021) image quantizer is trained with extra image data. VIM-Large is trained without dropout in transformers.

Table 7 shows the comparisons among different approaches divided into two categories: discriminative pretraining methods to distinguish among cropped or augmented image patches; and generative pretraining methods to generate image pixels or patches. The linear-probe accuracy of our proposed VIM with ViT-VQGAN are superior to other generative pretraining approaches like iGPT, and competitive with discriminative pretraining methods like BYOL (Grill et al., 2020) and DINO (Caron et al., 2021). Specifically, ImageNet-pretrained VIM-L significantly improves over iGPT-L, increasing linear-probe accuracy from 60.3% to 73.2% for a similar model size. VIM-L also outperforms iGPT-XL, which is larger and trained with extra web image data. Moreover, we also compare different stage 1 quantizers including CNN-based VQGAN and pretrained DALL-E dVAE (trained with extra web-scale image data)[2] in Table 7; these results are all worse than ViT-VQGAN quantizer, suggesting the importance of the multiple changes defined in Section 3.

## 6 ETHICS

Tasks involving generation raise a number of issues that should be considered, such as possible biases in underlying models and data—especially with respect to capabilities for people with different demographic backgrounds. The three datasets used in this paper–ImageNet, CelebA-HQ, and FFHQ–are all widely used in the literature, but it is worthwhile highlighting their unique natures and recent scholarship around them.

The FFHQ dataset[3] contains 70,000 images collected from Flickr, all of which have licenses appropriate for sharing, and the data maintainers provide means for individuals to opt-out of inclusion in the dataset. FFHQ was specifically collected to cover a broad range of demographics with respect to faces of people. This is confirmed in our models' generated examples, which cover a broad range of perceived ages, genders and ethnicities. Nevertheless, Balakrishnan et al. (2020) provide an extensive analysis of multiple forms of bias in datasets (including CelebA-HQ and FFHQ) and algorithms for face generation; not only do they find imbalances in skin tone in FFHQ, but also correlations between multiple attributes such as skin tone and hair length. Based on this and other

---

[2]https://github.com/openai/dall-e
[3]https://github.com/NVlabs/ffhq-dataset

factors such as privacy and copyright, they argue that synthetically-created face datasets, for which multiple attributes can be controlled, is an important direction of investment and general inquiry.

The CelebA-HQ dataset covers celebrities, which brings a consequent bias toward images of attractive people who are mostly in age range of twenty to forty years old. Esser et al. (2020) discusses these biases in details, and they furthermore project images from the FFHQ dataset onto CelebA-HQ: the main effect of which is to produce images of younger people with features conforming more to norms of celebrities popular in the United States of America. Our model's generations appear to have a similar bias as derived from training on CelebA-HQ. Neverethless, they do show broad coverage of different perceived genders and ethnicities, but with age skewed to the 20-40 year old range.

ImageNet is, of course, quite pervasive in computer vision. In this paper, we learn to generate images given ImageNet class labels; these labels mostly concern animals, plants and things. People are sometimes generated when conditioning on classes such as `sunglasses` since the training data images contain people wearing sunglasses, but the generated images contain few depictions of people overall. Nevertheless, it is important to recognize that ImageNet itself was created with biases in terms of image selection and label annotation as a result of its process of creation (Denton et al., 2021). Given this, results present on ImageNet cover a significant, but nonetheless biased, sample of the kinds of scenes and objects one might encounter across the entire world.

There are also potential problematic aspects of image generation models, as demonstrated with biases found in the PULSE model (Menon et al., 2020) (see Section 6) and in model correlations with human biases found in social psychology (Steed & Caliskan, 2021), as well as with possible uses of such models to create fake media (Westerlund, 2019).

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

## A    LINEAR-PROBE ON IMAGENET

We show linear-probe accuracy from different layers in a pretrained VIM-Base Transformer model in Figure 3. Similar to iGPT (Chen et al., 2020a), we also find the last few layers may not be the best layers for discriminative features, as the generative pretraining objective is to recover the original image tokens. The linear-probe accuracy increases quickly from the first transformer output, reaches its peak at middle layers, and finally decreases for the last few blocks. Interestingly, we find for both VIM-Base and VIM-Large, the middle transformer block has the near-best result. This observation connects the transformer model to an encoder-decoder model where the encoder encodes image tokens into high-level semantic features and the decoder takes feature information to generate output image tokens. We leave for future study regrading the interpretability of pretrained VIM models.

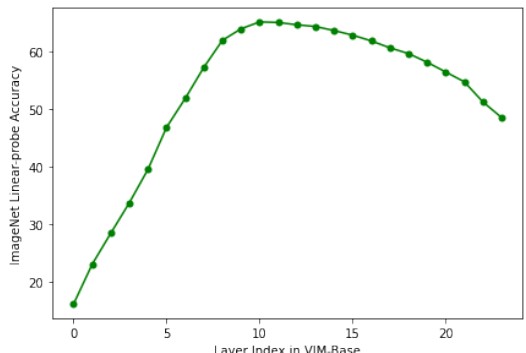

Figure 3: Linear-probe accuracy from different layers in a pretrained VIM-Base Transformer model.

## B    MODEL SIZES OF CLASS-CONDITIONED IMAGENET SYNTHESIS

We also present results of different sizes of Stage 2 Transformers for class-conditioned image synthesis and compare with VQGAN (Esser et al., 2021)[4] summarized in Table 8.

| Model | Stage-2 Transformer Size | #Tokens | FID | IS |
|---|---|---|---|---|
| Validation data | - | - | 1.62 | 235.0 |
| VQGAN (Esser et al., 2021) | 1.4B | $16 \times 16$ | 17.04 | 70.6 |
| **ViT-VQGAN + VIM-Base** | 650M | $16 \times 16$ | 11.20 | 97.2 |
| **ViT-VQGAN + VIM-Large** | 1.7B | $16 \times 16$ | 5.3 | 149.9 |
| **ViT-VQGAN + VIM-Base** | 650M | $32 \times 32$ | 8.81 | 110.8 |
| **ViT-VQGAN + VIM-Large** | 1.7B | $32 \times 32$ | 4.17 | 175.1 |

Table 8: FID comparison for class-conditional image synthesis on ImageNet with different Transformer sizes in Stage 2. Results are reported without rejection sampling.

## C    IMPLEMENTATION DETAILS OF FACTORIZED CODEBOOK

As we introduced in Section 3.2, we use a linear projection to reduce the encoded embedding to a low-dimensional variable space for code lookup. A detailed illustration is shown in Figure 4.

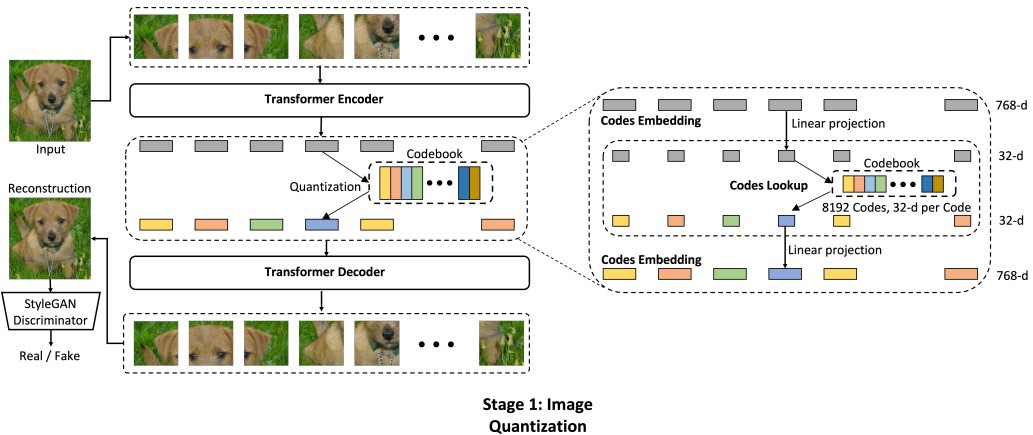

Figure 4: Illustration of factorized codes and codebook details.

## D    MORE SAMPLES ON CLASS-CONDITIONED IMAGENET SYNTHESIS

---

[4]https://github.com/CompVis/taming-transformers

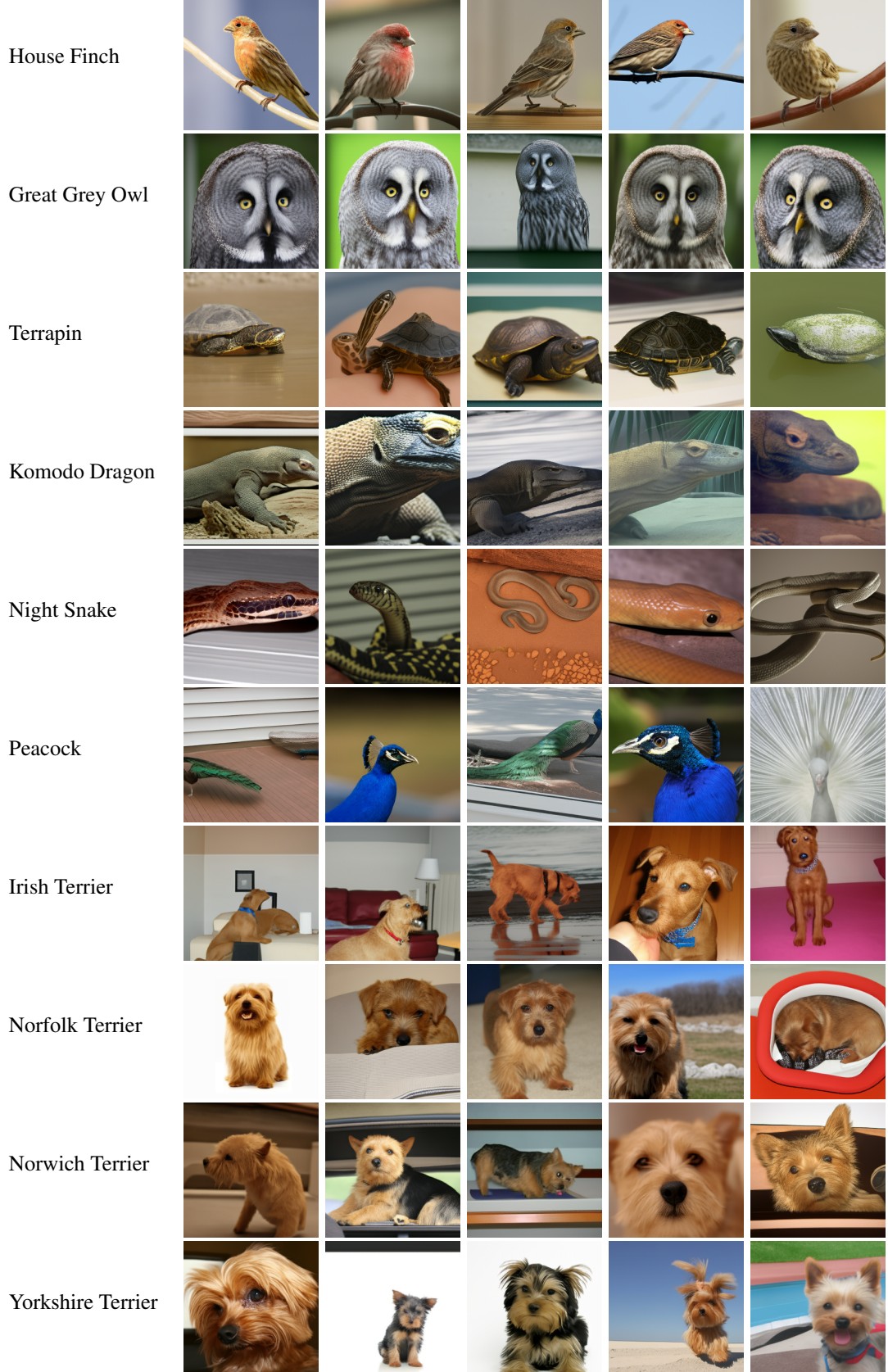

House Finch

Great Grey Owl

Terrapin

Komodo Dragon

Night Snake

Peacock

Irish Terrier

Norfolk Terrier

Norwich Terrier

Yorkshire Terrier

Figure 5: Uncurated set of samples from class-conditioned generation trained on ImageNet.

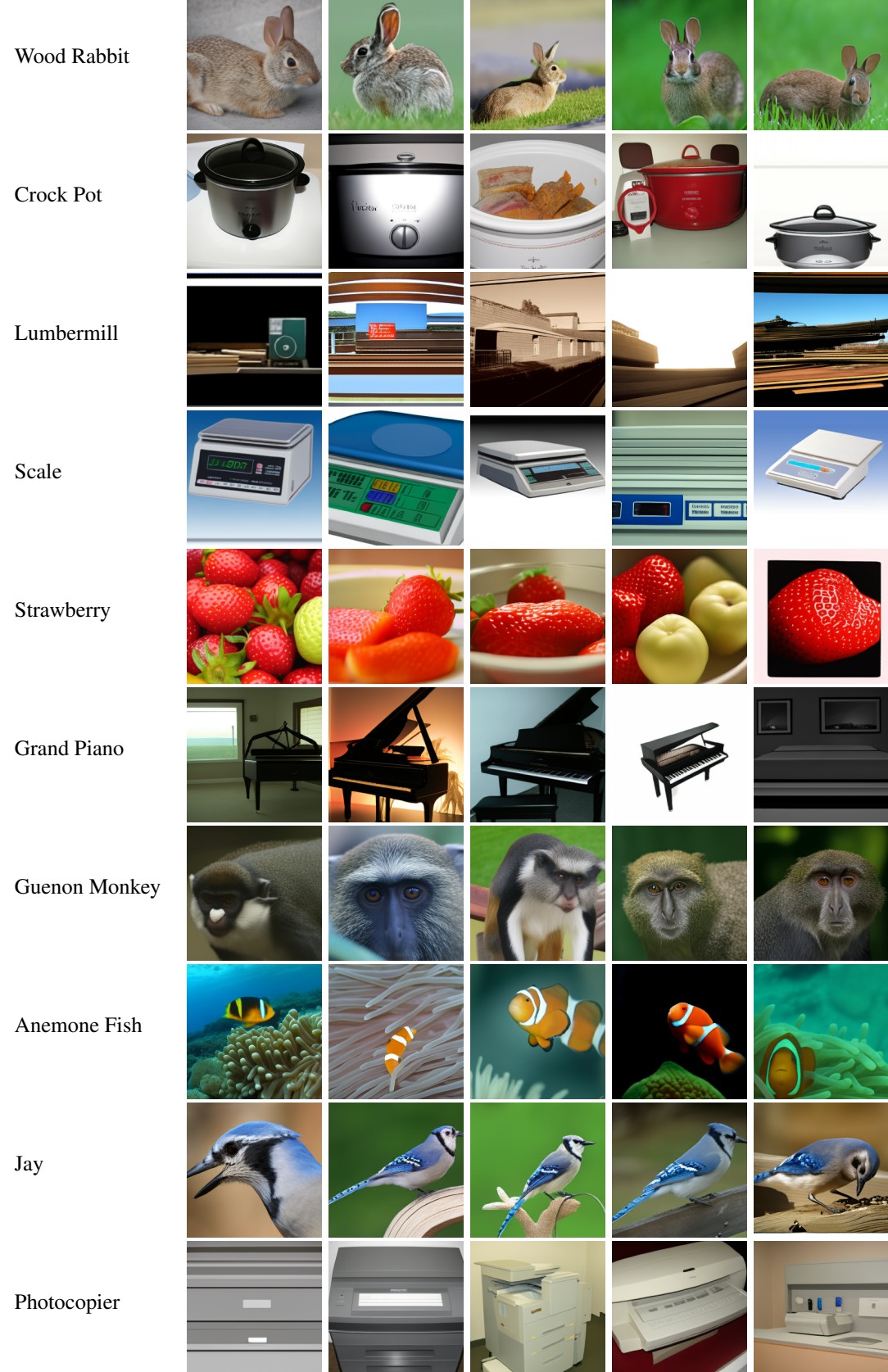

Figure 6: Uncurated set of samples from class-conditioned generation trained on ImageNet.

