# OpenReview forum: "Vector-quantized Image Modeling with Improved VQGAN"
_ICLR.cc/2022/Conference — ICLR 2022 Poster_

### Official Review · Reviewer_mxmU · 2021-10-30

**Correctness:** 3
**Technical Novelty And Significance:** 2
**Empirical Novelty And Significance:** 3
**Recommendation:** 6
**Confidence:** 4

**Main Review:**

strengths
========
- VQVAE and VQGAN have shown impressive generative performances but they were still lagging behind CNN based GAN models such as styleGAN. This work reduces the gap between them and shows qualitatively good generations.
- Although the model is a relatively straight-forward combination of ViT and VQGAN, its simplicity makes the model widely applicable
- The tricks introduced to improve the model such as projection to low dimensional space for code index lookup and l2-normalization of codes are useful findings.
- Strong quantitative results and ablation studies make the method convincing.
- The paper is written clearly.

weaknesses / questions
==================
- The claim for better codebook usage can be better supported by doing an analysis on their usage.
- In Section 3.3 multiple losses are used for training ViT-VQGAN. It would be good to see some quantitative results for choosing such a weighting between different losses, and some intuition of why the logit-laplace loss contributes to better codebook usage.
- For VQGAN results, is it using the same number of codebook entries as ViT-VQGAN? I could not find this information.
- As ViT has shown better performance than CNN based encoder/decoder, a natural next question is using it for the discriminator also. Have you tried this, if not, why not?
- Regarding styleGAN discriminator, was the improvement in metrics due to the discriminator architecture or the training procedure of stylegan discriminator (e.g its regularization)?
- Typo: the second last row in page 6, table 4 -> table 3

**Summary Of The Paper:**

This paper proposes vision transformer based VQGAN whose encoder and decoder are implemented as transformers rather than standard CNNs. It provides few improvements that demonstrates quantitatively better results in multiple datasets in unconditional / conditional image generation. It also shows how the proposed model can perform well as an unsupervised representation learning framework.


**Summary Of The Review:**

I believe this paper introduces a simple and effective method for improving VQGAN using vision transformers. The findings in this paper and the model would be useful to the community.

---

> ### Author Response · Authors · 2021-11-22
> **Thanks and Authors' Reply to Reviewer mxmU**
>
> Thanks for your time reviewing and providing detailed suggestions for our manuscript! We have updated the manuscript according to your suggestions and addressed all concerns below.
>
> ## Detailed Questions / Comments / Suggestions
>
> 1. The claim for better codebook usage can be better supported by doing an analysis on their usage.
>
> Thanks for the good suggestions! It is indeed informative to provide codebook usages. We have included the codes usage in Table 4 Ablation Study, and noted that "the codebook usage is calculated as the percentage of used codes (8192 codes in total) given a batch of 256 test images averaged over the entire test set".
>
> 2. In Section 3.3 multiple losses are used for training ViT-VQGAN. It would be good to see some quantitative results for choosing such a weighting between different losses, and some intuition of why the logit-laplace loss contributes to better codebook usage.
>
> Thanks for the suggestion! We'd be happy to provide more analysis of loss weighing experiments among adversarial loss, perceptual loss, logit-laplace loss and L2 loss in our final manuscript. Regarding why the logit-laplace loss contributes to better codebook usage, one intuition might be that “logit-Laplace loss can be viewed as normalized L1 loss which assumes the noise at the pixel level is laplace-distributed, while L2 loss assumes the noise is of a Gaussian distribution”. By having both losses, the quantizer needs to leverage more codes to have enough capacity thus higher codebook usages.
>
> 3. For VQGAN results, is it using the same number of codebook entries as ViT-VQGAN?
>
> Thanks for the question and apologize for the confusions! There are multiple tables that include VQGAN results. For Table 3, the number of codebook entries are provided in the table. For Table 4 ablation study and Table 7 linear-probing results, we compare quantizers of the same number of 8192 codebook entries. For Table 6 on class-conditioned ImageNet synthesis, we take the best results from VQGAN paper (16384 codebook entries in this case). We have updated the manuscript to be more clear!
>
> 4. As ViT has shown better performance than CNN based encoder/decoder, a natural next question is using it for the discriminator also. Have you tried this, if not, why not?
>
> Thanks for bringing up this discussion! We indeed had some initial experiments using ViT-based discriminator and we observed our results are slightly worse. We did not extensively study ViT-based discriminator. From many prior published works like StyleGANs, it seems discriminator architectures are more important in training GANs --- sometimes a change in activation function could lead to a huge difference (e.g., LeakyReLU with beta=0.2). To ensure fairness, in Table 4 we present CNN-based quantizers that are also trained with the same StyleGAN discriminator, the results are worse than ViT-VQGAN quantizers, validating the effectiveness of our proposed improvements.
>
> 5. Regarding styleGAN discriminator, was the improvement in metrics due to the discriminator architecture or the training procedure of stylegan discriminator (e.g its regularization)?
>
> Thanks for the question! We adapted both StyleGAN discriminator and its regularization. We explored experiments on StyleGAN-related changes (which are not presented in this manuscript as they are not very related to our main contributions), and found without R1 regularizer the results are considerably worse.
>
> 6. Typo: the second last row in page 6, table 4 -> table 3
>
> Thanks for the good catch! We appreciate it and have updated the manuscript accordingly!

---

> > ### Comment · Reviewer_mxmU · 2021-12-02
> > **Thanks for the reply**
> >
> > I sincerely thank the authors for answering my questions. I keep my recommendation as 6 as I believe this paper's simple effective method that extends VQGAN would be useful to the community

---

### Official Review · Reviewer_GySC · 2021-11-03

**Correctness:** 4
**Technical Novelty And Significance:** 3
**Empirical Novelty And Significance:** 3
**Recommendation:** 6
**Confidence:** 5

**Main Review:**

The performance gains, validated on different datasets, are very clear over state-of-the-art generative models, even compared with recent denoising diffusion methods. Meanwhile, the proposed modifications are simple and apparently effective. The paper is also well and clearly written.

Some other questions:
- Could you please give some explanations on ResNet-101 classifier-based rejection sampling? Is it the one used in VQVAE2 [1] and better than top-k sampling or nucleus sampling?
- What is the intuitive understanding to explain why  $\ell_{2}$ normalization is so vital?
- Is the generation quality only bounded by reconstruction? If not, what are the underlying flaws of auto-regressive probability models?
- Considering the fixed generation order of vector-quantized image modeling, I wonder if the new proposed reconstruction method  could combine with other transformer-based models like BERT, to further advance image inpainting tasks like [2]?

[1] Generating Diverse High-Fidelity Images with VQ-VAE-2, NeurIPS 2019.

[2] High-Fidelity Pluralistic Image Completion With Transformers, ICCV 2021.

**Summary Of The Paper:**

This paper mainly investigates how to further improve the image generation quality of previous work VQGAN, where several modifications are proposed to train a better quantized auto-encoder model. After modeling the discrete tokens with an auto-regressive transformer, we could observe that the generation results of the proposed ViT-VQGAN are really amazing and beat most of previous works in various benchmarks. Meanwhile, ViT-VQGAN also demonstrates its exceptional representation capabilities through unsupervised linear-probing.

To summarize, it seems there are several key factors to obtain a better reconstruction model:

- Project the latent codes into lower dimension, which leads to a larger codebook size and less ‘dead’ codes.
- StyleGAN Discriminator.
- $\ell_{2}$ normalization on latent variables.
- Stronger backbone (ViT-like architecture or decoder with more parameters).



**Summary Of The Review:**

Overall, the value of this paper is mostly experimental, or more like “bag of tricks”. The detailed analysis and explanation of why some components are essential are not stated very clearly, which may make the whole paper more interesting, but it indeed shines some light on the image generative model. Hence currently I tend to accept this paper.

---

> ### Author Response · Authors · 2021-11-22
> **Thanks and Authors' Reply to Reviewer GySC**
>
> ## Part 1: Main Concerns
>
> Q: Clear motivations and technical novelty.
>
> A: We first thank the reviewer for your insightful comments and questions, as well as encouraging comments tending to accept. We totally understand this concern and want to make it more clear as below (*mostly the same as our reply to Reviewer 6eTH in case you may miss it; our relies are based on reverse chronological order*).
>
> In our work, we aimed to explore the limit of vector-quantized image modeling in terms of both Image Generation and Image Understanding, and observed training a good image quantizer is the key. With our proposed improvements on image quantizer, we demonstrated superior results on both generation and understanding with the vector-quantized image modeling approach. We hope our results could be an encouraging milestone to motivate future work towards a more unified vector-quantized image modeling approach for both image generation and recognition.
>
> ## Part 2: Detailed Questions / Comments / Suggestions
>
> 1. Explanations on ResNet-101 classifier-based rejection sampling? Is it the one used in VQVAE2?
>
> Yes it is indeed the one used in VQVAE-2 presented in Section 3.3. The motivation of ResNet-101 classifier-based rejection sampling is to use a pre-learned classifier to re-score the sampled images based on predicted probability to the correct class. For example, the classifier-based rejection sampling with acceptance rate 0.5 means: given each class-id we randomly sample two images A and B; we get classification scores of this class-id for both image A and B; we only keep either image A or B with the higher classification score.
>
> 2. What is the intuitive understanding to explain why ℓ2 normalization is so vital?
>
> Thanks for the question! Our intuitions are that:
> - L2-norm maps all latent variables (codes) on a bounded hypersphere.
>
> - The Euclidean distance of l2-normalized latent variables l2($z_e(x)$) and l2($e$) evolves to the cosine similarity distance. It stabilizes the training and eases the optimization, sharing same merits as weight-norm [5].
>
> - There could be difficulty in optimization of discrete VQ with straight-through estimator, which could make l2-norm more important.
>
> - Strong empirical results are shown in ablation Table 4.
>
> [5] Salimans, Tim, and Durk Kingma. "Weight normalization: A simple reparameterization to accelerate training of deep neural networks." NeurIPS 2016.
>
> 3. Is the generation quality only bounded by reconstruction? If not, what are the underlying flaws of auto-regressive probability models?
>
> Thanks for the good question! The overall generation quality depends on both first-stage quantizer, and second-stage auto-regressive probability models (e.g.Transformers or other architectures). Transformers are de facto choices in the second stage, thus the majority of our improvements are on first-stage quantizer, where the quality depends on both reconstruction fidelity and the learned codebook quality. In this work, we show that our proposed improvements from architecture to codebook learning can lead to better reconstruction in first-stage quantizer (Table 3 and Table 4), better codebook usage, and overall quality in terms of both image generation (Table 5 and Table 6) and image understanding (Table 7).
>
> 4. Combine with other transformer-based models like BERT, to further advance image inpainting tasks like [2]?
>
> Yes we also believe it could be a next step to further exploring a BERT-like approach with vector-quantized image modeling, with the strongest image tokenizer like ViT-VQGAN. The resulting model may perform very well for both image understanding as well as image-inpainting tasks. We leave it to the research community for collaborative explorations together!

---

### Official Review · Reviewer_zYPH · 2021-11-03

**Correctness:** 3
**Technical Novelty And Significance:** 3
**Empirical Novelty And Significance:** 3
**Recommendation:** 6
**Confidence:** 4

**Main Review:**

**Strengths**
* The manuscript is well written, and easy to follow.
* Validating the discriminative power of autoregressive models on  ViT-VQGAN is a big plus. This shows the potential of generative models on classification tasks, just as in GPT.

**Weaknesses**
* In my opinion, comparing ViT-VQGAN of 32x32 latent dimension with VQGAN of 16x16 latent dimension in Table 6 is not fair at all, because the former is much slower than the latter in terms of sampling and training speed. This is my major concern in the experiments.
* It seems that ViT-VQGAN marginally improves VQGAN (1.28 vs. 1.49) in terms of rFID. Can we say that this improvement is significant? In addition, I’m still struggling with the reason why ViT is better than CNN in VQ methods. Is capturing the global relationship between patches (as in the classification) significant in VQ methods?
* It seems that the reproducibility is somewhat limited, because the detailed description on the computing environment is not presented. If the standard GPU machines having multiple V100 or A100 GPUs are used to train the 1.7B transformer with a sequence length of 1,024, it may not be possible to compute the gradient, even in the case of the single sample (batch size=1). I guess that the authors have tried many techniques, including mixed-precision training (obviously the standard technique), model sharding, gradient compression, storing Adam statistics in fp16 format, and so many other techniques. Unfortunately, I cannot find these details in the manuscript.

**Additional comments**
* The uncurated samples in Figure 2 and others in the supplementary are too smooth, which typically appears when we’re sampling with low top-k or top-p. Are these samples obtained by top-p=1.0 and top-k=8192?
* The reference to Table 3 is missing. Considering the context, the sentence “Table 4 shows FID between reconstructed images and original images ...” should be revised to refer to Table 3.

**Summary Of The Paper:**

This work replaces the CNN encoder/decoder into ViT to minimize the distortion caused by quantization as minimal as possible. Experiments show that transformers learned on the discrete representation learned by ViT-VQGAN can generate high-quality samples, and provide discriminative features better than iGPT.

**Summary Of The Review:**

* I’m seeing the potential and practical benefits of the proposed method, even though it is a fairly straightforward approach by replacing CNN into ViT in the VQ-GAN framework. In addition, including the linear probing experiments is also a plus.
* However, I have many concerns on the experiments and reproducibility. Without resolving these issues, my recommendation is borderline reject.

== Updates after the rebuttal ==

My major concerns have been addressed by the response, thereby raising my score from 5 to 6. I really appreciate the authors reporting these additional results in such a short time.

Though the empirical results of ViT-VQGAN are quite impressive, I still have a concern about the technical novelty. In the response to Reviewer 6eTH, the authors have argued that ViT-VQGAN is the first approach to consider vector quantization image modelling on both generation and recognition, while iGPT didn’t consider the vector quantization. However, according to the original iGPT paper (http://proceedings.mlr.press/v119/chen20s/chen20s.pdf) published in ICML’20 proceedings, they have reported the results of linear probing of iGPT on the top of discrete codes from VQ-VAE. Please revise the response to Reviewer 6eTH.

---

> ### Author Response · Authors · 2021-11-22
> **Thanks and Authors' Reply to Reviewer zYPH**
>
> Thanks for your time reviewing and providing detailed suggestions for our manuscript! We have updated the manuscript according to your suggestions and addressed all concerns below.
>
> ## Part 1: Main Concerns
>
> Q: Fairness of experiments: comparing ViT-VQGAN of 32x32 latent dimension with VQGAN of 16x16 latent dimension in Table 6 is not fair.
>
> A: Thanks for raising this concern! We present multiple experiments across different datasets in Table 3, 5 and 6, as well as a detailed ablation study on proposed modifications in Table 4. On ImageNet generation, we notice VQGAN uses 16x16 latent dimension while we use 32x32 (for simplicity we presented a single config of 32x32 latent dimension for all experiments across our manuscript).
> We also have results using our approach but with a 16x16 latent dimension in the table below. We hope those results can address concerns on a fair comparison to VQGAN on class-conditioned ImageNet generation and further validate the effectiveness of our presented techniques. Results have also been integrated into Supplementary Table 8 (across different sizes and latent dimensions) to ensure fair comparisons.
>
> |                       | Stage 2 Transformer Size | #Tokens |   FID |    IS |
> |-----------------------|--------------------------|---------|------:|------:|
> | VQGAN                 | 1.4B                     | 16x16   | 17.04 |  70.6 |
> | ViT-VQGAN + VIM-Base  | 650M                     | 16x16   |  11.2 |  97.2 |
> | ViT-VQGAN + VIM-Large | 1.7B                     | 16x16   |   5.3 | 149.9 |
> | ViT-VQGAN + VIM-Base  | 650M                     | 32x32   |  8.81 | 110.8 |
> | ViT-VQGAN + VIM-Large | 1.7B                     | 32x32   |  4.17 | 175.1 |
>
> ## Part 2: Detailed Questions / Comments / Suggestions
>
> 1. It seems that ViT-VQGAN marginally improves VQGAN (1.28 vs. 1.49) in terms of rFID. Can we say that this improvement is significant?
>
> Thanks for your question! The improvement of FID from 1.49 to 1.28 is a solid improvement, but we agree that when we look at reconstructed images, it’s hard to tell the visual differences. The results in Table 3 are only comparing stage-1 image quantizer and it does not capture stage-2 quality differences (which are presented in Table 5 and 6).
>
> 2. In addition, I’m still struggling with the reason why ViT is better than CNN in VQ methods. Is capturing the global relationship between patches (as in the classification) significant in VQ methods?
>
> Capturing the global relationship between patches are significant and we list a few evidences that may help understand this significance:
>
> - In VQGAN, the encoder is not a naive CNN-based encoder. A global relationship modeling module in the form of the non-local attention block is added into the CNN-based encoder. Earlier publication shows it significantly improves results [1].
>
> - The training of VQVAE/VQGAN is on unlabeled data and there are massive images available across the Internet. We don’t have to rely on inductive biases of convolution to learn good image encoders and decoders.
>
> - Thanks to Transformers’ computational efficiency and scalability, our empirical results in Table 4 show transformers outperform CNN in terms of fidelity-speed trade-offs.
>
> - It is important to keep a faster encoder because it speeds up Stage 2 training, where random augmentations are applied first to an image, followed by the encoder to obtain the input tokens.
>
> Given all the above reasons, we believe having ViT encoder-decoder in VQVAE/VQGAN is an important step and worth presenting our results to the research community.
>
>
> [1] Zhang, Han, et al. "Self-attention generative adversarial networks." ICML, 2019.
>
> 3. How to train the 1.7B transformer with a sequence length of 1,024? What techniques have authors tried?
>
> Thanks for the question! All our experiments are based on CloudTPUv4. To save memory, we use a factorized version of Adam, Adafactor [2], with the first moments quantized into Int8 [2] and a factorized second moments. We did not use mixed-precision training, model sharding, or gradient compression. We have updated the manuscript to include details in Section 5.2. A public implementation of Adafactor is available at https://optax.readthedocs.io/en/latest/api.html#optax.adafactor
>
> [2] Shazeer, Noam, and Mitchell Stern. "Adafactor: Adaptive learning rates with sublinear memory cost."  ICML 2018.
>
> 4. Are these samples obtained by top-p=1.0 and top-k=8192? Samples are smooth.
>
> Yes the images we present are without top-p/top-k sampling. Quantitative numbers like FID also measures diversity and qualitative  samples in Figure 2, 5, 6 also have a good diversity. If we understand the “smoothness” correctly, we think it also depends on the image classes (for example, “Komodo Dragon” in Figure 5 is less *smoother* than “Wood Rabbit” in Figure 6.)
>
> 5. The sentence “Table 4 shows FID between ...” should be revised to refer to Table 3.
>
> Thanks for the good catch! We appreciate it and have updated the manuscript accordingly!

---

> > ### Comment · Reviewer_zYPH · 2021-12-04
> > **Updates after the rebuttal**
> >
> > My major concerns have been addressed by the response, thereby raising my score from 5 to 6. I really appreciate the authors reporting these additional results in such a short time.
> >
> > Though the empirical results of ViT-VQGAN are quite impressive, I still have a concern about the technical novelty. In the response to Reviewer 6eTH, the authors have argued that ViT-VQGAN is the first approach to consider vector quantization image modelling on both generation and recognition, while iGPT didn’t consider the vector quantization. However, according to the original iGPT paper (http://proceedings.mlr.press/v119/chen20s/chen20s.pdf) published in ICML’20 proceedings, they have reported the results of linear probing of iGPT on the top of discrete codes from VQ-VAE. Please revise the response to Reviewer 6eTH based on iGPT version 1 paper (not version 2).

---

> > > ### Author Response · Authors · 2021-12-04
> > > **Thanks and Authors' Reply**
> > >
> > > Thanks for your reply!
> > >
> > > It's a good catch that iGPT reported a version with VQ-VAE in the later versions (but it seems iGPT didn't push the results with VQ-VAE further, compared with their main models with pixel color clustering). In Section 4.4, iGPT mentioned that without label ensemble, "we use the VQ-VAE data preprocessing step described in section 3.2, sacrificing local spatial invariance. Interestingly, when training on an IR of 192 × 192 × 3 and a MR of 48 × 48, the best-layer accuracy remains unchanged at 65.3%". This in fact confirms our contributions of this work that "in our work, we aimed to explore the limit of vector-quantized image modeling in terms of both Image Generation and Image Understanding, and observed training a good image quantizer is the key" in our responses.
> > >
> > > Interestingly, we were mainly referencing the paper pdf from OpenAI's official website (https://cdn.openai.com/papers/Generative_Pretraining_from_Pixels_V2.pdf) in which no VQ-VAE results are reported. We acknowledged these VQ-VAE results in iGPT and updated our comments with a note.

---

### Official Review · Reviewer_6eTH · 2021-11-06

**Correctness:** 4
**Technical Novelty And Significance:** 2
**Empirical Novelty And Significance:** 4
**Recommendation:** 6
**Confidence:** 4

**Main Review:**

This paper builds on top of VQGAN and DALL-E like models to achieve SOTA FID and IS scores for image modeling and unsupervised feature learning. This paper is well-written and mostly clear, however a few clarifications would be helpful (see below).

Strengths:
this paper achieves very good empirical results by combining various minor innovations based on first quantize the image representation followed by training a transformer based LM. The description of the architecture as well as the presentation of the results are very clear.

Cons:
While the paper is clearly written, there do needs to be a few classifications.
- For example, 'perceptual loss' is mentioned in several places in the paper but no definition or comment is made about what exactly is being used. It is better to have a footnote for the completeness of the paper.
- In the introduction, it isn't super clear what are the main novelty of the paper over VQGAN and DALL-E, which also uses quantization and transformers. It would be better if contributions are outlined more clearly, especially w.r.t. the architecture in DALL-E.
- Training time, hardware used are missing in section 5.1. How long did it take to train 500K steps, and what are the 128 accelerators?

Questions:
- for Eq. 1 in section 3.2, is it not equivalent to get rid of the stop-gradient operators and just have a single term in the loss function, but multiply /beta to the weights of the encoder during training?
- Are there implementation or code available accompanying this paper?


**Summary Of The Paper:**

This paper builds on top of VQGAN and DALL-E like models to achieve SOTA FID and IS scores for image modeling and unsupervised feature learning.

**Summary Of The Review:**

This paper achieves very good empirical results and build on-top of several recent works on VQ + Transformers for image modeling. However the paper lacks some details and clarifications. Given the significant improvement of FID and IS, the paper could also do more to analyze which components of the proposed method was instrumental in achieving these gains.

---

> ### Author Response · Authors · 2021-11-22
> **Thanks and Authors' Reply to Reviewer 6eTH**
>
> Thanks for your time reviewing and providing detailed suggestions for our manuscript! We have updated the manuscript according to your suggestions and addressed all concerns below.
>
> ## Part 1: Main Concerns
>
> Q: Outlined more clearly on contributions of this work compared with VQGAN and DALL-E.
>
> A: Thanks for the question on novelty over VQGAN and DALL-E, and suggestions to clearly list our contributions! Indeed we want to note that both VQGAN and DALL-E are built upon VQVAE [1]. VQGAN [4] uses GAN and perceptual loss to train quantizer, while DALL-E [3] uses logit-laplace loss. Both VQGAN and DALL-E only study image generation tasks, as shown in the Table we summarized below.
>
> In our work, we aimed to explore the limit of vector-quantized image modeling in terms of both Image Generation and Image Understanding, and observed training a good image quantizer is the key. With our proposed improvements on image quantizer, we demonstrated superior results on both generation and understanding with the vector-quantized image modeling approach. We hope our results could be an encouraging milestone to motivate future work towards a more unified vector-quantized image modeling approach for both image generation and recognition.
>
> |                  | Vector-quantized Image Modeling | Image Generation | Image Understanding (Recognition) |
> |------------------|:-----------------------------------------:|:----------------:|:---------------------------------:|
> | VQVAE [1], 2017  |                    ✔                    |        ✔       |                 **✘**                |
> | iGPT [2], 2020   |                     **✘**\*                    |        ✔       |                ✔                |
> | DALL-E [3], 2021 |                    ✔                    |        ✔       |                 **✘**                |
> | VQGAN [4], 2021  |                    ✔                    |        ✔       |                 **✘**               |
> | Ours (This Work)             |                    ✔                    |        ✔       |                ✔                |
>
> \* iGPT's main models are based on pixel color clusters. VQ-VAE encoder based results are reported (as mentioned by Reviewer zYPH) but results are no better than pixel-based. See our responses to Reviewer zYPH for more details.
>
> [1] Oord, Aaron van den, Oriol Vinyals, and Koray Kavukcuoglu. "Neural discrete representation learning." Preprint, 2017.
>
> [2] Chen, Mark, et al. "Generative pretraining from pixels." ICML, 2020.
>
> [3] Ramesh, Aditya, et al. "Zero-shot text-to-image generation." Preprint, 2021.
>
> [4] Esser, Patrick, Robin Rombach, and Bjorn Ommer. "Taming transformers for high-resolution image synthesis." CVPR, 2021.
>
>
> ## Part 2: Detailed Questions / Comments / Suggestions
>
> 1. 'perceptual loss' is mentioned in several places in the paper but no definition or comment is made.
>
> Thanks for the suggestion and we have revised the paper to include citations of perceptual loss in every place. The perceptual loss used in our work is no different than the common setup, which is the L2 distance on the linear-projected feature space of a pretrained VGG. A public implementation can be found in https://github.com/richzhang/PerceptualSimilarity/blob/master/lpips/lpips.py
>
> 2. How long did it take to train 500K steps, and what are the 128 accelerators?
>
> The training of 500K steps takes roughly 36 hours on 128 CloudTPUv4. Our ablation comparisons are all done under the same hardwares and software stacks, and can be replicated on other hardwares like GPUs. We have revised our paper to make them more explicit!
>
> 3. For Eq. 1 in section 3.2, is it not equivalent to get rid of the stop-gradient operators and just have a single term in the loss function, but multiply /beta to the weights of the encoder during training?
>
> It is indeed equivalent to multiplying $\beta$ into learning encoder (for example, learning rate). One caveat on multiplying $\beta$ is that the encoder (and the decoder) also receives gradients from reconstruction losses in Section 3.3, so we will need to update those losses accordingly. Overall having a formulation of current Eq. 1 is cheap and less error-prone.
>
> 4. Are there implementation or code available accompanying this paper?
>
> As we discussed in detail in Section 6 Ethics, tasks involving generations raise a number of issues that should be considered, such as possible biases in underlying models, weights and data, especially w.r.t. capabilities for people with different demographic backgrounds. The code of this work and the models we trained might be used for malicious purposes without proper licenses and agreements. While we are still working on scrutinizing potential malicious purposes and their broader impacts, to make a final decision if we release our code and models, we are happy to take any technical questions on implementation details that are not clear. *To all readers and reviewers, please feel free to leave any technical questions on OpenReview.*

---

> > ### Comment · Reviewer_6eTH · 2021-11-29
> > **Feedback to Author rebuttal**
> >
> > Thanks to the authors for comments addressing my concerns in the initial review, as well as updating the manuscript.
> > As a result, I'm reasonably satisfied and will be upgrading my recommendation score to 6 from 5. Overall, while this paper might not have ground breaking novelty, it is an interesting datapoint on achieving excellent empirical results in this domain.
> >
> > Given that it's not easy to reproduce results solely based on the paper, and given similar comments from another reviewer, it is strongly recommended for the authors to release the implementation that can reproduce these results.

---

### Public Comment · ~Yongfei_Liu1 · 2021-11-20
**The question about "Factorized Codes"**

Hi, author;
Recently, I have read your paper ViT-VQGAN, which is an excellent work. But I still have some problems, which confuses me a lot.

As you introduced, you use a linear projection to reduce the output of the encoder to a low-dimensional variable space for code lookup.
I am very confused How to interpret the "code lookup" and "code embedding", Are there two embedding matrices in vector quantization? or just one codebook matrix?   If there is just one codebook, how to make it as "code lookup" and "code embedding"?

Another question is that what's the dimension of $\mathcal{z}_e(x)$ and $e$? I guess they are both 32-d based on your description at Factorized codes.

Looking forward to your help. Thanks

---

> ### Author Response · Authors · 2021-11-21
> **Authors' Reply to the Reader's Question**
>
> Hi Yongfei,
>
> Thanks for the question, as well as acknowledging the excellence of this work!
>
> We apologize for the confusion due to the naming. You are correct on the dimension of $z_e(x)$ and $e$, both are 32-d vectors. In our work, we refer the "code embedding" as the encoded outputs from an input image, while "code lookup" as the projected low-dimensional codes ($z_e(x)$) to match a learnable codebook or the codes ($e$) in codebook.
>
> To avoid confusion to other readers, **we have added an illustration figure (Figure 4) in Appendix C** and hopefully it could be cleaner. Please let us know if you may have further questions!
>
> Best,
>
> Authors of ICLR 2022 Conference Paper488

---

### Decision · Program_Chairs · 2022-01-20

**Decision:**

Accept (Poster)

**Comment:**

This paper adopts ViT in the VQ-GAN framework replacing CNN, and achieves SOTA FID and IS scores. The empirical results are pretty impressive. It could benefit some practical applications.

The technical novelty is limited, but the tricks such as l2-normalization of codes are interesting.